# Changes in the Protein Composition of the Aqueous Humor in Patients with Glaucoma: An Update Review

**DOI:** 10.3390/ijms26073129

**Published:** 2025-03-28

**Authors:** Maria Kiełbus, Dominika Kuźmiuk, Aleksandra Magdalena Skrzyniarz, Aleksandra Zynkowska, Joanna Dolar-Szczasny, Tomasz Chorągiewicz, Robert Rejdak

**Affiliations:** 1Student Scientific Club at the Department and Clinic of General and Paediatric Ophthalmology, Medical University of Lublin, Chmielna 1 Str., 20-079 Lublin, Poland; 62111@student.umlub.pl (M.K.); 58231@student.umlub.pl (D.K.); 63302@student.umlub.pl (A.M.S.); 63906@student.umlub.pl (A.Z.); 2Department of General and Paediatric Ophthalmology, Medical University of Lublin, Chmielna 1 Str., 20-079 Lublin, Poland; tomasz.choragiewicz@umlub.pl (T.C.);

**Keywords:** glaucoma, aqueous humor, protein composition, open-angle glaucoma, angle-closure glaucoma

## Abstract

The study of the aqueous humor (AH) plays a key role in understanding the pathophysiology of glaucoma. The AH provides nutrition, maintains the appropriate intraocular pressure, and provides important information about the mechanisms of the disease. The development of modern technologies has allowed the use of more accurate analytical methods, which has proven to be a key factor in determining the changes occurring in the proteome of the aqueous humor of glaucoma patients. Recently, researchers have observed changes in the levels of proteins associated with inflammation, oxidative stress, the complement system, and extracellular matrix remodeling. They have also shown that these changes may be variable for different types of glaucoma. The objective of this review is to collect and summarize the current knowledge on the potential biomarkers and pathomechanisms involved in the pathogenesis of glaucoma. We hope that our review will contribute to the improvement of current diagnostic methods in this illness and, through a better understanding of the changes occurring during the progression of the disease, will enable the development of more effective preventive and therapeutic strategies in the future.

## 1. Introduction

The aqueous humor (AH) is a fluid that fills the anterior and posterior chambers of the eye [1,2,3,4]. It plays a crucial role in maintaining proper intraocular pressure and nourishing structures like the cornea and lens. However, in certain diseases, changes in the composition of the AH can contribute to the development of pathological conditions [5,6].

The term “glaucoma” refers to a group of diseases that includes some of the most common causes of complete vision loss worldwide. This disease is associated with the progressive atrophy of the optic nerve, which may be asymptomatic for a significant period of time. Consequently, late detection carries a higher risk of irreversible blindness [7,8]. However, the level of risk, symptoms, and nature of treatment are primarily influenced by the type of the disease [8].

Different types of glaucoma vary in their causes. In primary open-angle glaucoma, the main factors include elevated intraocular pressure, chronic oxidative stress, disturbances in the microcirculation of the optic nerve, and a genetic predisposition [1,5,6,8]. Primary angle-closure glaucoma results from anatomical predispositions of the eye, such as a shallow anterior chamber, a larger lens, and an increased iris thickness [4,6]. Secondary glaucoma may be a consequence of diabetes, intraocular inflammation, eye tumors, trauma, or the prolonged use of corticosteroids [3,6,8]. Congenital glaucoma is caused by the abnormal development of eye structures during the prenatal period, leading to a defective aqueous humor outflow from birth [6,7,8].

Regardless of the type of ailment, common features of glaucoma are the progressive degradation of the optic nerve with the destruction of its nerve fibers [7,9,10] and the pathological deepening and widening of the optic disc, as well as damage to the retinal ganglion cells and its neural layer [1,5]. These features result in changes in the field of vision and increased intraocular pressure [9,11].

The highest incidence of glaucoma is observed in highly developed countries [12]. Currently, glaucoma affects 2.1% of adults over 40 years of age, which means about 980 thousand patients [13]. Additionally, at least one million people struggle with ocular hypertension (OHT). Most cases of glaucoma—as many as two thirds—occur in people over 70 years of age. It is predicted that by 2060, due to the aging of the population, the percentage of patients will increase to 2.8% [14].

Primary angle-closure glaucoma (PACG) differs fundamentally from primary open-angle glaucoma (POAG), yet both represent the most prevalent types of glaucoma [15].

Early diagnosis and intervention are key to halting disease progression and preventing complications. Despite advances in diagnosis and treatment, the full understanding and effective management of glaucoma remain a clinical challenge [14].

The risk of glaucoma increases with age and is also dependent on gender—acute PACG is more common in women, and chronic PACG is more common in men [7,12,16]. Modern imaging techniques, such as ultrasound biomicroscopy (UBM) and optical coherence tomography (OCT), enable the precise assessment of the structures of the anterior segment of the eye, which allows for the better identification of patients at risk of developing glaucoma [16].

PACG is more likely to lead to blindness than POAG, indicating a poorer prognosis for this form of glaucoma [6]. PACG accounts for more than 50% of glaucoma-related blindness, especially in people of Asian descent. A specific form of this disease is primary acute angle-closure glaucoma (PAACG), which is a significant cause of vision loss in East Asia. Its prevalence increases with age and is often associated with cataracts, which further increases the risk of its development [3,6].

POAG is more common than PACG in Europe and North America [17]. However, it is particularly prevalent in the Caucasian population [18]. It is also the most common type in the United States [11]. It is estimated that 0.5–1.5% of people aged 40–80 years develop the disease over a 5-year period. The risk of developing open-angle glaucoma increases with age: in people aged 70–79 years, the risk is more than 12 times higher compared to those aged 40–49 years [18].

In contrast, in 2013, the number of people with PACG was 20.17 million, and projections indicate that it increased to 23.36 million in 2020 and may reach 32.04 million in 2040 [19].

To enhance accessibility and provide a clearer understanding of the study’s methodology, we have included a schematic illustration of the study flow (Figure 1). This visual representation aims to outline the key steps and processes involved, allowing readers to quickly grasp the structure of the research and follow the progression of the study more effectively.

## 2. Materials and Methods

A comprehensive search of the literature was conducted in English using the PubMed, ResearchGate, and Google Scholar databases, covering publications from 1 January 2006 to 31 December 2024. The search employed the following keywords: “proteome change in aqueous humor glaucoma”, “proteomic change in aqueous humor glaucoma”, “change in proteome in primary open-angle glaucoma”, “aqueous humor in different types of glaucoma” and “change in aqueous humor in angle-closure glaucoma”. Figure 2 details the screening workflow, illustrating the attrition of records from initial identification (n = 4691) to final inclusion (n = 144) with documented exclusion reasons. The initial search yielded a substantial number of results, which were screened for relevance based on titles and abstracts. Studies were included if they focused on the protein composition of the aqueous humor in glaucoma patients, provided original data (e.g., experimental studies, clinical trials), were published within the specified timeframe (2006–2024), and were written in English. The exclusion criteria included non-English publications and studies not directly related to glaucoma or aqueous humor proteomics. After applying the inclusion and exclusion criteria, a total of 144 articles were selected for the in-depth review. Reference management software (Zotero v. 7.0.15 was used to organize the retrieved articles and remove duplicates.

## 3. Pathomechanism

The aqueous humor is a fluid located in the anterior chamber of the eye that is crucial for maintaining intraocular pressure and nourishing the eye tissues [20]. It is produced daily by the ciliary body processes in the posterior chamber of the eye [21,22]. After fulfilling its functions, the aqueous humor is drained from the eye through the trabecular meshwork located in the iridocorneal angle or through the uveoscleral outflow into the episcelar space [20,21,22,23]. Therefore, if additional elements, such as blood, inflammatory cells, pigment cells, and other metabolic products, appear in the fluid, they block the sieve and the fluid cannot drain freely from the eye [24,25]. However, in glaucoma, the outflow is most often obstructed without any apparent cause [24,26]. The outflow process of the aqueous humor is as follows: after being drained through the pupil, it flows through the trabecular meshwork in front of the scleral spur and the iris implant to the Schlemm’s canal, then to the episcleral venous system, the orbital venous system, and finally, to the systemic venous circulation [21,23,24].

The impeded drainage of the AH results in elevated intraocular pressure, exerting stress on the optic nerve, which is widely considered to be a major factor contributing to the development of glaucoma [21,23,24,26]. This condition ultimately leads to the degeneration of nerve fibers and contributes to a gradual reduction in the visual field [15,22,27]. Changes observed in the eye reveal a concave or curved shape of the optic disc when examined during a fundus evaluation [24,26].

There are several types of glaucoma, depending on the cause and progression of glaucomatous neuropathy. A common feature is the characteristic loss of retinal ganglion cells (RGCs) and associated damage to the optic nerve [20,28].

The most common division of glaucoma, based on the mechanism of its development, is open-angle glaucoma (OAG) and angle-closure glaucoma (ACG). In the first case, the most common, there is a gradual blockage of the channels draining fluid from the inside of the eye. The increase in intraocular pressure is then small, and the progression of the disease is slow [15,22,24].

Risk factors for the development of open-angle glaucoma include advanced age, because over time there is a decrease in aqueous humor drainage through the trabecular meshwork, while aqueous humor production is only slightly reduced [21,26]. This imbalance leads to increased mean intraocular pressure and greater diurnal fluctuations, which are common in patients with glaucoma [29]. The development of this disease is also facilitated by a family history of glaucoma, previous eye trauma, vasospastic diseases (e.g., migraines), and reduced blood circulation in the optic nerve (which may be the result of low blood pressure or severe anemia) [15,26].

ACG is much less common and occurs when the angle of the eye becomes completely blocked in a short period of time [21]. This leads to a rapid increase in pressure in the eyeball, which results in clearly noticeable symptoms, such as a severe headache, photophobia, or profuse tearing, and may quickly lead to vision loss [15,24]. Typically, this type of glaucoma occurs in only one eye [27,30].

There is also a division in which glaucoma is divided depending on whether it develops in a previously healthy eye (primary) or occurs as a result of previous diseases (secondary) [15]. The causes of secondary glaucoma include, among others, lens lesions, the inflammation of the anterior uvea, ciliary-irido-lens occlusion (malignant glaucoma), the scarring of blood vessels (neovascular glaucoma, which is associated with diabetes and circulatory diseases), or canal obstruction caused by adhesions or neoplastic changes [15,24,27]. In rare cases, damage to the optic nerve may occur with normal intraocular pressure [21,24]. The cause of this damage is probably the disruption of the blood and oxygen supply to the optic nerve [24,26,30].

## 4. Methods for Determining the Protein Composition of the Aqueous Humor

The aqueous humor is directly in contact with the site of pathogenesis in glaucoma [3,15,20,22,25]. It performs the important function of providing nutrients while maintaining refraction and appropriate intraocular pressure [3]. Therefore, this fluid provides a lot of information regarding the pathophysiology of glaucoma [20]. An analysis of the protein composition of the aqueous humor aims to identify the biomarkers involved in the disease mechanism, improve glaucoma diagnostics, individualize treatment, and monitor the disease’s progression and its response to treatment [20,22].

Initially, the methods for determining protein composition were limited and were based mainly on differences in protein masses when comparing healthy and glaucomatous eyes. An example of this is a case-control study conducted by Zaidi et al. in 2010 [31]. In their study, 0.1 mL of aqueous humor from the anterior chamber was collected from the participants and analyzed by the Bradford method. The molecular weight of the proteins was analyzed using sodium dodecyl sulfate–polyacrylamide gel electrophoresis (SDS-PAGE). The study showed that the difference in the protein contents between the groups was statistically significant, but it did not provide information on the type of proteins detected [32].

Another method used to determine aqueous humor protein contents is the ELISA [32,33,34]. An example of its use is the study described by Tsutomu Igarashi et al. in 2021 [33], in which the ELISA was used to determine the level of brain-derived neurotrophic factor (BDNF) in the aqueous humor of patients with glaucoma. It provided important clinical trial data that were used to develop a gene therapy for the eye.

ELISA was also used in a study by Nore Woltsche et al. in 2023, where neurofilament light chain (NfL) levels in serum and the aqueous humor were measured [34]. It showed that the NfL levels in the anterior chamber fluid were elevated in the patients with glaucoma and correlated with the intraocular pressure and the retinal nerve fiber layer thickness [34]. These data confirmed that the aqueous humor NfL levels might be used as a new marker of glaucoma, supporting the effectiveness of applying ELISA in the diagnosis of this disease [20,35,36].

Currently, research is based on the precise identification of the type of molecules that occur and undergo changes in the aqueous humor of the diseased eye and not on the determination of their masses alone [20,36]. Mass spectrometry-based studies introduce innovative methods that effectively cope with the challenges resulting from a limited sample volume and a low protein content [20,35]. The most commonly used method, due to its high sensitivity and specificity, is liquid chromatography–mass spectrometry (LC-MS). This method combines the chromatographic separation of samples and the analysis of their compositions by measuring the ratio of the mass of charged molecules to their charge [35]. It offers high specificity, especially in detecting different isoforms or post-translationally modified forms of proteins, which allows for a precise understanding of the differences in aqueous humor compositions between healthy and glaucomatous eyes [36]. 

Another example of a method used is the quantitative analysis of the aqueous humor proteome using hyper-response monitoring mass spectrometry (HRM-MS), based on the Sequential Window Acquisition of all Theoretical Mass Spectra (SWATH) technology [22,37,38]. This is a PCR-based method that detects methylation levels at specific sites, and an appropriately selected primer structure ensures a high test sensitivity [37,38]. The ability to detect just a few copies of methylated DNA makes this technique very important in establishing links between exposure to environmental factors, epigenetic changes, and disease. This method was used, for example, in 2016, by Kaeslin et al. to analyze the AH proteome in patients with POAG and a control group [22]. The study identified 448 proteins and found 87 proteins that showed differences in the aqueous humor of the people with glaucoma compared to the healthy people.

The constantly developing diagnostic methods for analyzing the protein composition of the aqueous humor, combined with the increasing availability of modern technologies, contribute to the deepening understanding of the mechanisms of this complex neurodegenerative disease [20,26].

## 5. Protein Composition of Aqueous Humor in Different Types of Glaucoma

Recent advances in proteomic analysis have provided valuable insights into the molecular alterations characteristic of different glaucoma subtypes. Table 1 offers a systematic review of studies conducted between 2006 and 2024, focusing on changes in the aqueous humor proteome. Meanwhile, Table 2 presents comparative data on protein signatures between glaucoma and cataract cohorts. Together, these findings highlight disease-specific perturbations, revealing distinct proteomic profiles in glaucoma patients and offering potential avenues for targeted diagnostic and therapeutic strategies.

### 5.1. Primary Open-Angle Glaucoma

Numerous studies have focused on the proteomic profile of the AH in patients with glaucoma; predominantly those involved in POAG diagnosis have revealed notable alterations in the concentrations of the proteins associated with the complement system, oxidative stress, inflammation, and the remodeling of the extracellular matrix (ECM) [22,51,54].

Studies by Kaeslin et al. (2016) [22] and Mok et al. (2024) [54] have shown changes in cholesterol metabolism through the increased synthesis of apolipoproteins, such as APOA1, APOA4, APOC1, and APOD [51]. There is evidence that the overexpression of apolipoproteins contributes to cellular damage and apoptotic changes, as well as causing excessive oxidative stress that leads to the activation of plasminogen and the remodeling of the extracellular matrix [3,56]. This also suggests that apolipoproteins present in the aqueous humor may increase IOP by increasing outflow resistance [54,56]. An exception is the increased synthesis of the apolipoprotein APOD, which has been confirmed to be negatively involved in various neurological disorders, including glaucoma, by inhibiting apoptosis and oxidative damage in patients with POAG [51,57].

Research indicates that changes in glucose metabolism also occur in the course of POAG [22,46]. The downregulation of, among others, pyruvate kinase, alpha enolase, and triosephosphate isomerase indicates a reduced ability of the pathologically changed tissue to carry out the glycolysis process, which confirms disorders in carbohydrate metabolism and a change in the energy demand by degenerated cells in the course of glaucoma [22,46,56].

Protein composition changes in the AH in POAG also have a major impact on oxidative-stress-related processes [22,46,54,58]. There is a decrease in the expression of proteins involved in hydrogen peroxide reduction, such as catalase and peroxiredoxins 1, 2, and 6 [22]. In addition, decreased levels of pyruvate kinase, L-lactate dehydrogenase A chain, ubiquitin-60S ribosomal protein L40, transforming growth factor beta-2, aldehyde dehydrogenase, and pleiotrophin suggest a weakened ability of glaucoma patients to adapt to oxidative stress [22,54]. The upregulation of L-methionine sulfoxide, an oxidized form of methionine, in diseased tissues often leads to impaired biological functions, partly due to its role in antioxidant production for glutathione [46,54].

Significant changes in protein composition also remarkably affect neurodegenerative processes in glaucoma [22,46,51]. Nerve cells show increased sensitivity to elevated levels of oxidative stress induced by reactive oxygen species (ROS). This process is associated with a decrease in antioxidant defense mechanisms in the aqueous humor, which consequently leads to neuronal damage [59]. The downregulation of heat shock protein beta-1 (HSPB1), phosphoglycerate kinase 1 (PGK1), phosphatidylethanolamine-binding protein 1 (PEBP1), and neuroserpin (SERPINI1) indicates disorders of neurogenesis, axonal cell growth, and synaptic plasticity [22].

Another group of proteins detected in POAG patients in the study conducted by Nikhalashree et al. (2019) were calsequestrin and sarcalumenin, which contribute to neurodegeneration through calcium-related mechanisms [46]. There are also studies indicating a significant increase in the concentration of afamin (AFM), which, unlike other proteins affecting the cells of the nervous system identified in the AH in patients with POAG, is a glycoprotein exhibiting neuroprotective properties by binding vitamin E and having antioxidant effects [51,59].

The increased expression of pigment epithelium-derived factor (PEDF), Dickkopf-related protein 3 (DKK3), and Wnt inhibitory factor 1 (WIF1) glycoproteins [51] is also worth noting. These proteins play a crucial role in Wnt signaling pathways, which regulate axonal maturation, remodeling, and the maintenance of homeostasis, ensuring the proper functioning of mature neurons [51,60]. The appropriate regulation of Wnt complex proteins, therefore, plays a significant role in protecting nerve cells from neurodegeneration [61].

It is also believed that the upregulation of PEDF, DKK3, and WIF1 glycoproteins may be associated with adaptive processes in response to increased intraocular pressure in the course of glaucoma, as demonstrated in the study conducted by Guo et al. (2019) [45].

Researchers have demonstrated an association between the levels of secreted frizzled-related protein-1 (SFRP1), a protein that antagonises Wnt signaling pathways, and intraocular pressure in POAG patients. They found that POAG patients with high IOP had lower levels of secreted SFRP1 than patients with normal IOP [45]. In their study, a significantly higher level of transforming growth factor-β2 (TGF-β2) in the aqueous humor was also observed in the patients with POAG than those with other types of glaucoma.

Opinions on the participation of individual components of the complement system and proteins involved in its regulation in the course of POAG are divided [53,54]. In the study by Mok et al. (2024), a significant increase in the concentrations of proteins such as C3, C8a, and vitronectin (VTN) was observed in the AH in patients with POAG compared to patients with cataracts [54]. VTN is a strong activator of the complement system, which finds its use by binding to plasminogen activator inhibitor 1 (PAI-1) and regulating changes occurring in the ECM [3].

In opposition, Hubens et al. (2021) published the results of a study on the changes in the ratios of complement factors C3a to C3 in the aqueous humor and serum [53]. They divided the patients into three groups: patients with progressive POAG (group 1), patients with stable POAG (group 2), and patients with cataracts (control group). While the concentration of C3 in serum and the AH did not differ significantly between the groups, they showed that the C3a factor in serum and the AH was significantly increased in group 1. They argued that the C3a/C3 ratio of the complement system in both the aqueous humor and serum correlates with the severity of POAG, and the obtained results indicate that complement activation plays a key role in the progression of glaucoma and is variable depending on the stage of the disease [53].

### 5.2. Primary Angle-Closure Glaucoma

The proteome analysis of the aqueous humor conducted in 2019 by Kaur et al. in patients with PACG revealed significant changes in the organization of the actin cytoskeleton [48]. The dysfunction of the proteins regulating its structure leads to the formation of cross-linked actin networks (CLANs), which reduces the elasticity of the trabecular meshwork and impairs the outflow of the aqueous humor, contributing to increased intraocular pressure [62,63]. Additionally, an increase in oxidative stress was observed in PACG patients, which was caused by the reduced activity of antioxidant enzymes, such as superoxide dismutase (SOD) and glutathione peroxidase (GPx), and a decrease in the total antioxidant capacity (TAC) [48,64,65].

In another study conducted in the same year by Adav et al. [49], patients with primary angle-closure glaucoma showed the presence of atypical collagens and fibronectins and the decreased activity of 72 kDa collagenase IV, which leads to the accumulation of extracellular matrix components and increased intraocular pressure [4,49]. An increase in matrix metalloproteinase-16 (MMP16) was also observed, which indicates disorders in ECM degradation [66,67]. Additionally, the decreased expression of proteins key to neuronal processes, such as neural precursor cell-expressed, developmentally downregulated protein 4 (NEDD4) and T-complex protein 1 subunit theta (CCT8), and increased levels of antioxidant proteins, including catalase, were found. These changes disrupt ECM homeostasis and neuronal functions, contributing to disease progression and vision loss [49,64,66].

In 2021, an analysis of the aqueous humor of the eyes of dogs with PACG was carried out [52]. During the study, a significantly higher protein concentration was found compared to the healthy control group—osteopontin (SPP1) increased, the level of which was 81 times higher in the PACG study group, which indicates its role in the inflammatory processes of the trabecular meshwork and increased intraocular pressure [68]. The second key protein was vimentin (VIM), the level of which increased 8.9-fold, which is associated with damage to the retinal nerve fiber layer and healing mechanisms [69,70]. The results indicate that SPP1 and VIM may be key in the pathogenesis of this disease [52,68,69,70].

Other studies, however, have focused on the role of inflammatory cytokines. In the analysis conducted by Feng et al. in 2024, the concentration of IL-4 in the aqueous humor was significantly higher in patients with CPACG [55]. Thus, the results suggest that IL-4 may be involved in the development of the CPACG mechanism, although its increase does not directly affect neurodegeneration or visual functions [55,71]. Increased concentrations of IL-36, IL-37, and IL-38 were also demonstrated in the aqueous humor of patients with CPACG [50,72]. A significant correlation was observed between these cytokines and the mean visual field deviation (MDVF), suggesting their involvement in the immunological pathogenesis of CPACG. IL-36, associated with the inflammatory response, has been previously detected in acute uveitis, and its role in the progression of CPACG may result from the activation of immune cells [73]. IL-37 and IL-38, which are anti-inflammatory cytokines, probably play a compensatory role in the ocular microenvironment, indicating the non-infectious, inflammatory nature of CPACG [50,74,75,76].

### 5.3. Secondary Glaucoma

Secondary open-angle glaucoma is a complex group of conditions characterized by elevated intraocular pressure due to impaired aqueous humor outflow in an open angle [10,77,78]. Various pathologies leading to an impaired outflow may include the deposition of pseudoexfoliative material, pigment accumulation in the trabeculation, ocular trauma, neovascularization, and chronic steroid use [10,77,78]. In contrast to primary open-angle glaucoma, in which the mechanism of the disease is not fully understood, the secondary forms of glaucoma have clearly defined causes [77].

#### 5.3.1. Pseudoexfoliation Glaucoma

Pseudoexfoliation syndrome (PEX) is one of the most common causes of secondary open-angle glaucoma [79,80]. The cause of PEX is the accumulation of protein–glycosaminoglycan deposits in the aqueous humor. These deposits interfere with the function of the outflow structure, leading to an increase in intraocular pressure [79,80,81,82,83]. Elevated levels of the proteins associated with extracellular matrix metabolism, such as lysyl oxidase (LOXL1) and matrix metalloproteinases (MMP-2, MMP-9), are present in the aqueous humor of patients with PEX [66,81,84]. Polymorphisms in the LOXL1 gene have been associated with a predisposition to the development of this disease [81,84,85,86,87]. Oxidative stress, aging, and genetic predisposition play a key role in the pathogenesis of pseudoexfoliation glaucoma [42,44,81,84]. In addition, the presence of growth factors, including TGF-β2, promotes fibrosis, which worsens the aqueous flow [83,84,85].

#### 5.3.2. Neovascular Glaucoma

Neovascular glaucoma develops as a result of the proliferation of pathological blood vessels in the glaucoma angle, leading to angle closure and a significant increase in intraocular pressure. The disease is most commonly associated with diabetic retinopathy, central retinal vein thrombosis, or other conditions that lead to retinal ischemia [88,89,90]. Significantly elevated levels of VEGF (vascular endothelial growth factor), which plays a key role in inducing angiogenesis, have been found in the aqueous humor of patients with neovascular glaucoma [90,91]. VEGF stimulates the formation of new abnormal blood vessels in the angle that are prone to damage and bleeding. In addition, elevated levels of platelet-derived growth factor (PDGF) promote fibrosis and worsen the permeability of the angle of infiltration [89]. The elevated levels of pro-inflammatory cytokines, such as IL-6 and TNF-α, observed in the aqueous humor of patients with neovascular glaucoma suggest an important role for chronic inflammation in the pathogenesis of this disease [41,47,92]. Liu et al. demonstrated that these cytokines may further exacerbate the damage to the trabecular meshwork by stimulating inflammatory cell migration and increasing oxidative stress [3].

#### 5.3.3. Pigmentary Open-Angle Glaucoma

Pigmentary open-angle glaucoma often develops in younger patients, particularly those with pigment dispersion syndrome (PDS). In the pathomechanism, the pigment from the iris epithelium is released into the aqueous humor and deposited in the trabecular meshwork, where it obstructs the fluid outflow [78,93,94,95]. Advanced imaging studies have shown that patients with PDS often have deeper anterior chambers and more concave irises, which predisposes them to pigment release during normal eye movements [93,96,97,98]. Research has also shown that markers of oxidative stress, such as superoxide dismutase (SOD) and catalase, are elevated in the aqueous humor of people with PDS. This suggests that oxidative damage plays a central role in the progression of the disease [96,97]. Migliazzo et al. demonstrated in their long-term study that persistent pigment deposition in the trabecular meshwork can lead to chronic damage and progressive visual field loss in patients with pigmentary dispersion syndrome and pigmentary glaucoma [99]. Recent research points to the potential for specific treatments to reduce oxidative stress and inflammation as adjuncts to traditional IOP-lowering therapies, offering a more complete strategy for managing this challenging condition.

#### 5.3.4. Steroid-Induced Open-Angle Glaucoma

The chronic use of steroids, both systemic and topical, can lead to steroid-induced open-angle glaucoma. Glucocorticosteroids reduce the permeability of the trabeculum by the deposition of collagen and the remodeling of the extracellular matrix [100,101,102]. Patients with this form of glaucoma have been found to have elevated levels of transforming growth factor beta-2 (TGF-β2) in the aqueous humor. Elevated levels of TGF-β2 in the aqueous humor promote the fibrosis of the trabeculum and the excessive deposition of extracellular matrix proteins, which restrict the aqueous humor outflow [101,102,103]. Proteome studies have also revealed the increased activity of matrix metalloproteinases (MMP-2 and MMP-9) and decreased levels of the tissue inhibitor of metalloproteinases (TIMP-1), creating an imbalance that further disrupts trabecular meshwork function. These biochemical changes are thought to be influenced by dysregulated glucocorticoid receptor signaling, which modifies how ocular cells react to prolonged steroid exposure [39,101,104]. This points to the complexity of the disease and indicates the value of screening. Research into alternative therapies has indicated promise in preventing irreversible damage to the optic nerve, including selective IOP-lowering agents and the early cessation of steroid use [105,106]. The timely adjustment or discontinuation of steroids is essential to reduce the risk of irreversible optic nerve damage.

#### 5.3.5. Post-Traumatic Glaucoma

Ocular trauma is another important cause of secondary open-angle glaucoma. Post-traumatic glaucoma can develop as a result of mechanical damage to the structures responsible for maintaining intraocular pressure (IOP). When the trabecular meshwork is disrupted, fluid accumulates, and pressure builds up within the eye [107,108,109,110,111]. In research, trauma-induced inflammatory responses, including elevated levels of markers such as lactate dehydrogenase (LDH), interleukin-1β (IL-1β), and tumor necrosis factor-alpha (TNF-α), are major contributors to this process. These markers point to a cascade of immune responses that not only exacerbate the damage but also contribute to the scarring of the trabecular meshwork, further impairing fluid drainage. The extracellular matrix changes and trauma-induced fibrosis can create long-term complications, making it harder to control IOP [40,43,112,113]. Early anti-inflammatory and anti-fibrotic therapies could decrease the risk of permanent optic nerve damage.

### 5.4. Pathways Implicated in Glaucoma

One of the principal signaling pathways implicated in glaucoma is the apoptotic pathway [114]. Apoptosis, commonly referred to as programmed cell death, has been identified as a contributing factor in the degeneration of retinal ganglion cells (RGCs), which are essential for relaying visual information to the brain in cases of glaucoma. Research indicates that increased intraocular pressure (IOP) and oxidative stress—two significant contributors to glaucoma progression—can trigger apoptotic signaling pathways, ultimately resulting in RGC death [64,115]. Another signaling route implicated in glaucoma is the oxidative stress pathway [65,114,115,116,117,118,119,120,121,122,123,124]. Oxidative stress can activate signaling pathways that contribute to cellular damage and apoptosis. In the context of glaucoma, it has been associated with the progressive degeneration of retinal ganglion cells, while antioxidant treatments have demonstrated protective effects in certain animal models of this condition [116,121,122,125,126,127]. Inflammation represents an additional signaling pathway that plays a role in the progression of glaucoma [128,129]. Inflammation within the eye has the potential to initiate signaling pathways that result in the degeneration of RGCs and the advancement of the condition. Research indicates that increased IOP and oxidative stress can trigger pro-inflammatory signaling mechanisms in glaucoma, which subsequently promote the secretion of cytokines and other signaling factors, ultimately leading to cell damage and death [130,131].

Several proteins are critical in the pathways implicated in glaucoma. Among these, the myocilin protein, encoded by the MYOC gene, is one of the most extensively studied. Researchers have linked mutations in this gene to increased IOP and the development of POAG [132]. It is widely accepted that myocilin contributes to the maintenance of the structural integrity and function of the trabecular meshwork, a tissue that regulates aqueous humor outflow [133,134]. The influence of the outflow on cellular adhesion and extracellular matrix composition is well documented. The abnormal accumulation of misfolded myocilin can lead to cellular stress and apoptosis, contributing to increased IOP [134,135,136].

Another key protein, optineurin, has been associated with both familial and sporadic forms of glaucoma. Researchers hypothesize that optineurin plays a role in neuroprotection and the regulation of inflammatory responses in the optic nerve head [137,138]. Studies have shown that mutations in the OPTN gene, which encodes optineurin, can disrupt its normal function, increasing the susceptibility to glaucomatous damage [139,140]. Additionally, proteins such as transforming growth factor-beta and matrix metalloproteinases have been implicated in the remodeling of the extracellular matrix (ECM) within the trabecular meshwork [141,142]. TGF-β signaling has been shown to induce fibrosis and elevate outflow resistance, while MMPs, through their action of degrading ECM components, fulfil a dual regulatory function in maintaining tissue homeostasis and contributing to disease progression [143,144].

## 6. Conclusions

Researchers investigating the protein changes in the aqueous humor have provided important insights into the glaucomatous pathogenesis. While there are many similarities between the proteins in different types of glaucoma, there are also some differences. Common changes include the proteins involved in inflammation, oxidative stress, the complement system, and extracellular matrix remodeling. These changes reflect the many factors that contribute to glaucoma, such as problems with AH drainage, high intraocular pressure, and nerve damage. However, the studies vary in their results, suggesting that more consistent research is needed to determine whether these protein changes can be used as reliable biomarkers to help diagnose and classify glaucoma.

Several studies have found higher levels of apolipoproteins in primary open-angle glaucoma, suggesting that these proteins may play a role in oxidative stress and ECM remodeling. This may lead to increased resistance to AH outflow, raising IOP. Studies also show alterations in glucose metabolism and antioxidant enzymes, such as lower levels of catalase and peroxiredoxin. This adds to the role of oxidative stress in glaucoma. The role of the complement system in POAG is not fully understood. Studies have shown conflicting results. Some studies, like that by Mok et al. (2024), show that complement proteins, like C3 and C8a, are activated, while other studies, such as that by Hubens et al. (2021), suggest that the activation varies depending on the stage of the disease [53,54]. These differences indicate that these results should be confirmed in larger studies.

Regarding primary angle-closure glaucoma, studies have shown that the balance of the ECM is disturbed. There is an increased expression of cross-linked actin networks (CLANs) and abnormal collagen types, which reduce the flexibility of the trabecular meshwork and block the aqueous humor outflow. Inflammatory cytokines, such as IL-4 and IL-36, are also associated with PACG, suggesting that inflammation plays a role in the disease. These cytokines have been associated with clinical measures, such as visual field loss, suggesting that they may be useful as biomarkers. However, more research is needed to understand whether these cytokines are directly involved in nerve damage or vision loss.

Proteomics is an important area that can serve to improve glaucoma care. We can learn more about how the disease works and find new ways to diagnose and treat it by understanding the protein changes in the aqueous humor. Future research should aim to include larger patient cohorts, employ standardized techniques more consistently, and explore how protein levels evolve as the disease progresses. These steps are essential if proteomics is to become a reliable tool in the diagnosis and treatment of glaucoma.

## 7. Limitations

Although this review provides a comprehensive analysis of changes in the aqueous humor protein composition in patients with glaucoma, several limitations must be considered when interpreting the findings.

The studies included were selected based on searches of major scientific databases, applying the specific exclusion criteria described earlier. However, a selection bias remains a concern, as unpublished data, non-English studies, and those not indexed in commonly used databases were excluded.

A major limitation of many proteomic studies is their small sample sizes, often with fewer than 10 patients per group. Such small cohorts reduce the statistical power, limiting the generalizability of findings and making it difficult to identify strong biomarkers or therapeutic targets. Additionally, the interindividual variability in aqueous humor protein composition—especially across different glaucoma subtypes and disease severities—may not be fully captured.

Methodological differences also present a challenge, particularly in sample preparation and proteomic techniques. While advanced approaches, such as liquid chromatography–mass spectrometry (LC-MS) and hyper-reaction monitoring mass spectrometry (HRM-MS), offer greater sensitivity and accuracy, they had not been widely used across all the studies, leading to inconsistencies in the reported protein changes.

Another key limitation is the predominance of cross-sectional data, which hinders the ability to distinguish whether changes in protein composition are a cause or a consequence of glaucoma progression. Longitudinal studies tracking protein alterations over time would be crucial for establishing causal relationships and identifying reliable biomarkers.

While proteomic studies have proposed several potential biomarkers for glaucoma, their clinical utility remains uncertain. Few differentially expressed proteins have been extensively validated in large patient cohorts, and further research is needed to determine their diagnostic, prognostic, and therapeutic potential.

## Figures and Tables

**Figure 1 ijms-26-03129-f001:**
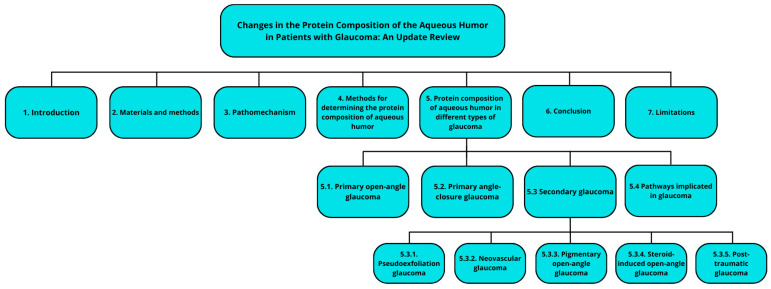
Schematic illustration of the study flow.

**Figure 2 ijms-26-03129-f002:**
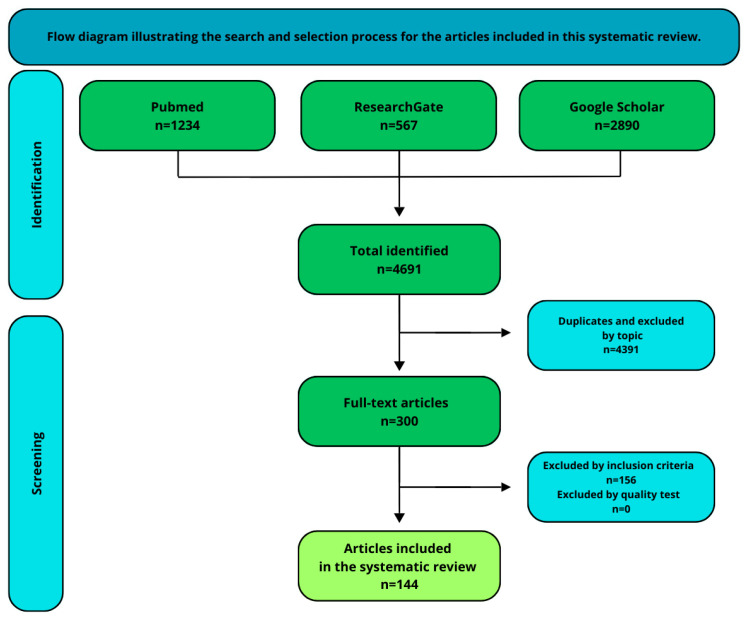
Flow diagram illustrating the search and selection process for the articles included in this systematic review.

**Table 1 ijms-26-03129-t001:** Selection of studies examining changes in the aqueous humor proteome between 2006 and 2024.

Authors	Type of Study	Year	Country	Study Group	Control Group	Patient Number
Määttä et al. [39]	Analytical study on MMP and TIMP levels and activation	2006	Finland	URSG	POAG, cataract	13 URSG 10 POAG 10 cataract
Balaiya et al. [40]	Observational study	2011	USA	POAG	cataract	32 POAG 32 cataract
Ohira et al. [41]	Cross-sectional study with cytokine analysis	2015	Japan	NVG, POAG	cataract	33 NVG 36 POAG 68 cataract
Dursun et al. [42]	Observational study with biochemical analysis	2015	Turkey	PEX syndrome, PEG	cataract	26 PEX syndrome 26 PEG 26 cataract
Bojikian et al. [43]	Retrospective, observational case series	2015	USA	OGI	-	515 OGI
Kaeslin et al. [22]	Case-control study with quantitative proteomic analysis	2016	Switzerland	POAG	cataract	5 POAG 5 cataract
Tetikoğlu et al. [44]	Prospective clinical study with biochemical analysis	2016	Turkey	PEX syndrome	healthy subjects	34 PEX syndrome 38 healthy subjects
Guo et al. [45]	Cross-sectional, observational study	2019	China	POAG, CACG, PACS	cataract	21 POAG 19 CACG 9 PACS 24 cataract
Nikhalashree et al. [46]	Comparative, observational study with proteomic analysis	2019	India	POAG, PACG	cataract	90 POAG 72 PACG 78 cataract
Sun et al. [47]	Observational study with biomarker analysis	2019	China	NVG, PDR	-	15 NVG 17 PDR
Kaur et al. [48]	Comparative, cross-sectional study with proteomic analysis	2019	India	POAG, PACG, cataract	-	9 POAG 9 PACG 9 cataract
Adav et al. [49]	Comparative, cross-sectional proteomics study	2019	Singapore	PACG	cataract	2 PACG and cataract 3 cataract only
Zhang et al. [50]	Comparative, observational study	2019	China	CPACG	cataract	22 CPACG 29 cataract
Liu H et al. [51]	Experimental study with proteomic analysis	2020	Germany	POAG	cataract	23 POAG 12 cataract
Liu X et al. [3]	Observational study with proteomic analysis	2021	China	PAACG, PCACG, NVG, cataract	-	57 PAACG 50 PCACG 35 NVG 33 cataract
Yun et al. [52]	Comparative, observational study with proteome analysis	2021	Republic of Korea	PACG	healthy subjects	6 PACG 6 healthy subjects
Hubens et al. [53]	Comparative observational study	2021	Netherlands	progressive POAG, stable POAG	cataract	10 progressive POAG 10 stable POAG 10 cataract patients
Mok et al. [54]	Multi-omics analysis	2024	Republic of Korea	POAG, XFS, XFG	cataract	5 XFS 4 XFG 11 POAG 7 cataract
Feng et al. [55]	Comparative, cross-sectional, observational study	2024	China	CPACG	cataract	31 CPACG 30 cataract

PAACG—primary acute angle-closure glaucoma. PCACG—primary chronic angle-closure glaucoma. NVG—neovascular glaucoma. POAG—primary open-angle glaucoma. PACG—primary angle-closure glaucoma. PACS—primary angle-closure suspects. XFS—exfoliation syndrome. CACG—chronic angle-closure glaucoma. XFG—exfoliation glaucoma. PEX—pseudoexfoliation syndrome. PEG—pseudoexfoliative glaucoma. PDR—proliferative diabetic retinopathy. OGI—open-globe injuries. URSG—uveitis-related secondary glaucoma. CPACG—chronic primary angle-closure glaucoma.

**Table 2 ijms-26-03129-t002:** Selection of comparative studies with proteomic analysis and numbers of proteins detected in aqueous humor in patients with glaucoma and cataract.

Authors	Year	Country	Patient Number	Cataract Proteins (Total Number)	Glaucoma Proteins (Total Number)	Comment
Kaeslin et al. [22]	2016	Switzerland	5 POAG 5 cataract	448	448	Proteins different for both groups: 87 (in glaucoma, 34 were upregulated and 53 were downregulated).
Nikhalashree et al. [46]	2019	India	90 POAG 72 PACG 78 cataract	184	POAG 190 PACG 299	Unique proteins for cataract: 97; POAG: 87; PACG: 171. Proteins identified in every group: 58. Proteins identified in POAG and PACG: 43.
Kaur et al. [48]	2019	India	9 POAG 9 PACG 9 cataract	636	POAG 594 PACG 625	Unique proteins for cataract: 221; POAG: 206; PACG: 226. Proteins identified in every group: 246. Proteins identified in POAG and PACG: 55.
Liu H et al. [51]	2020	Germany	23 POAG 12 cataract	175	175	No data for exact number of proteins upregulated or downregulated in glaucoma AH (6 proteins were described as with “significant up-regulation”).
Liu X et al. [3]	2021	China	57 PAACG 50 PCACG 35 NVG 33 cataract	315	315	Differential proteins PAACG/cataract: 31. Differential proteins PCACG/cataract: 17. Differential proteins PCACG/cataract: 100. Differential proteins PAACG/PCACG: 202.
Mok et al. [54]	2024	Republic of Korea	5 XFS 4 XFG 11 POAG 7 cataract	329	329	Differential proteins XFS/cataract: 15. Differential proteins XFG/cataract: 34. Differential proteins POAG/cataract: 51.

PAACG—primary acute angle-closure glaucoma. PCACG—primary chronic angle-closure glaucoma. POAG—primary open-angle glaucoma. PACG—primary angle-closure glaucoma. NVG—neovascular glaucoma. XFS—exfoliation syndrome. XFG—exfoliation glaucoma.

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
