# Peer review of "Changes in the Protein Composition of the Aqueous Humor in Patients with Glaucoma: An Update Review"

_ijms, 2025, doi:10.3390/ijms26073129_

Round 1
Reviewer 1 Report
Comments and Suggestions for Authors
The manuscript contains important aspects. Several comments are recommended to make the manuscript have more impact.
Table 1 should contain subject number as the study number is missing in the this table.
If it is available, it is good to add pathway analyses for those selected proteins to find important pathways managing retinal diseases. This is because the manuscript only describes the protein in each section. But it is important to know what kinds of pathways are involved in the disease condition. This part is missing in this manuscript.
Countries info should be further added in each table. Only author name and year are provided in the tables. It is important to know about the place that conducts the experiment. It can give us more comprehensive understanding.
Schematic illustration to summarize the study should be added. As a review paper, it is good to summarize the study as one image so that the future readers can follow up the study flow easily.
In the method section, it is written that PubMed, ResearchGate, and Google Scholar databases are used for the study. How those databases are selected and used should be clearly presented with each search's number with illustrations.
It might be important to introduce disease progression in the intro. How glaucoma is made and how it is processed long termly should be presented in the introduction.
Limitations should be well-presented and discussed as they are missing in this article.
--------------------------------------------------
The manuscript contains important aspects. Several comments are recommended to make the manuscript have more impact.
Table 1 should contain subject number.
If it is available, it is good to add pathway analyses for those selected proteins to find important pathways managing retinal diseases.
Countries info should be further added in each table.
Schematic illustration to summarize the study should be added.
In the method section, it is written that PubMed, ResearchGate, and Google Scholar databases are used for the study. How those databases are selected and used should be clearly presented with each search's number with illustrations.
It might be important to introduce disease progression in the intro.
Limitations should be well-presented and discussed.
Reviewer 2 Report
Comments and Suggestions for Authors
General comments
The manuscript ‘Changes in the Protein Composition of the Aqueous Humor in Patients with Glaucoma: An Update Review’ by Maria Kiebus et al. comprehensively reviewed the current knowledge of protein markers in the aqueous humor (AH). The authors covered the basic biology of AH, assessment techniques for protein quantification and described the different types of glaucoma. The manuscript can serve to provide updated information regarding the specified topic.
However, the presentation demands extensive revision. Much redundancy and ambiguity have to be clarified/avoided.
Specific comments
- Introduction, paragraph 2: suggested to revise as-- The term "glaucoma" refers to a group of diseases that constitute among the most common causes of complete vision loss worldwide.
- Introduction, paragraph 2: suggested to revise as--This disease is associated with progressive atrophy of the optic nerve…….
- Introduction, paragraph 3: suggested to revise as--common features of glaucoma are:…….
- Introduction, paragraph 3, line 4: suggested to revise as—These features result in changes in the field of vision and increased intraocular pressure [9,11].
- Introduction, paragraph 4, line 2: ‘….age, which means about 980 thousand patients.’ This needs a reference citation. Also, in the last paragraph of Introduction, ‘ In contrast, in 2013, the number of people with PACG was 17 million, and projections indicate that it has increased to 23.36 million in 2020 and may reach 32.04 million in 2040.’ This also needs a reference citation. The number of patients is quite different in the two descriptions.
- Materials and methods, lines 2 -3: from January 1, 2006 to when? This needs to be defined.
- Materials and methods, line 3: ‘Initially, we used…….’. Does this mean the key words have been changed later in this research? If not, delete the word.
- Pathomechanism, paragraph 1, first sentence: ‘Aqueous humor (AH)’ has been defined in the Introduction. The authors are suggested to use only ‘Aqueous humor’ or ‘AH’.
- Pathomechanism, paragraph 1, line 4: The common consensus has been the existence of two outlets for aqueous humor drainage, including the trabecular meshwork and the uveoscleral outflow. It is suggested that the authors also put the uveoscleral outflow here.
- Pathomechanism, paragraph 2: ‘…… elevated intraocular pressure…(line1); ‘Intraocular pressure (IOP) elevation…’ (Iine 3). Suggested to put these together.
- Pathomechanism, paragraph 2, last sentence: suggested to revise as—(which may be the results of low blood pressure and severe anemia).
- Methods for determining the protein composition of aqueous humor, paragraph 2, line 4: suggested to revise as— ‘collected from the participants and analyzed by the Bradford method.’
- Methods for determining the protein composition of aqueous humor, paragraph 3: suggested to revise as—‘Another method used to determine aqueous humor protein contents is the ELISA [31,32,33].’
- Methods for determining the protein composition of aqueous humor, paragraph 4: suggested to revise as—‘These data confirmed aqueous humor NfL might be used as a new marker of glaucoma, supporting the effectiveness of applying ELISA in the diagnosis of this disease [19,34,35]’.
- Methods for determining the protein composition of aqueous humor, paragraph 5: suggested to revise as—‘combines chromatographic separation of samples and analysis of their compositions by measuring the ratio of….’.
- Methods for determining the protein composition of aqueous humor, paragraph 5: ‘……..or post-translationally modified forms of proteins [9a],….’. What is the reference [9a]?
- Methods for determining the protein composition of aqueous humor, paragraph 5: suggested to revise as—‘which allows for precise understanding the differences in aqueous humor compositions between the healthy and the glaucomatous eyes.’
- Methods for determining the protein composition of aqueous humor, paragraph 6: suggested to revise as—‘The ability to detect few copies of methylated DNA makes this technique very important….’.
- In Table 1, The type of study: Analytical study on MMP and TIMP levels and activation, cross-sectional study with cytokine analysis. They should start either all capital or non-capitals. The same applies to Table 2.
- Primary open-angle glaucoma, paragraph 1, line 2: suggested to revise as—‘predominantly those involved in POAG diagnosis and …..’.
- Primary open-angle glaucoma, paragraph 2: ‘An exception is the increased synthesis of apolipoprotein APOD, which has been confirmed to be negatively involved in various neurological disorders, …..’ .
- Primary open-angle glaucoma, paragraph 4: ‘Upregulation of methionine L-sulfoxide in diseased tissues, which is an oxidised form of methionine, often leads to impaired biological functions through the production of an antioxidant for glutathione [45,53].’ This entire sentence is ambiguous and confusing.
- Primary open-angle glaucoma, paragraph 5: ‘…..elevated oxidative stress induced by reactive oxygen species (ROS)’. ‘ ….indicates disorders of neurogenesis, axonal cell growth and synaptic plasticity [21].’.
- 3.4. Steroid-induced open-angle glaucoma, last sentence: ‘Emphasise the urgency of stopping steroids or adjusting the dose in a timely manner to reduce the risk of irreversible damage to the optic nerve.’. This sentence is grammatically wrong and should be recast.
- Conclusion: ‘Researches into the protein changes in the aqueous humor have provided important insights into the glaucomatous pathogenesis’.
The above-mentioned are just some examples of ambiguity and redundancy. It is highly recommended that the authors seek assistance from a professional English editor.

The presentation demands extensive revision. Much redundancy and ambiguity have to be clarified/avoided.
Round 2
Reviewer 1 Report
Comments and Suggestions for Authors
Well-addressed all. No further issue.
Reviewer 2 Report
Comments and Suggestions for Authors
The manuscript has been substantially improved in writing and presentation, with errors corrected.